# Affecting Factors of Prostate Volume in Forensic Autopsied Decedents

**DOI:** 10.3390/healthcare11101486

**Published:** 2023-05-19

**Authors:** Kota Tanaka, Masahito Hitosugi, Marin Takaso, Mami Nakamura, Arisa Takeda

**Affiliations:** Department of Legal Medicine, Shiga University of Medical Science, Otsu 520-2192, Japan; ds111950@g.shiga-med.ac.jp (K.T.); marint@belle.shiga-med.ac.jp (M.T.); mamin@belle.shiga-med.ac.jp (M.N.); arisa204@belle.shiga-med.ac.jp (A.T.)

**Keywords:** prostate, volume, age, cardiovascular disease, cerebrovascular disease, lifestyle-related disease

## Abstract

Because decedents undergoing forensic autopsies would have behaved normally before death, prostate volume according to age group can be confirmed with forensic autopsy materials. The objectives of this study were to first confirm the current prostate volume by age and then determine diseases that can influence prostate volume using forensic autopsy materials. Prostate specimens were collected from forensic autopsies performed at Shiga University of Medical Science, Japan, between January 2015 and December 2019. Overall, 207 decedents were included in the study. Prostate volume was measured by the Archimedes’ principle. Concomitant diseases were determined by the past medical histories and autopsy results. The mean crude prostate volume was 29.1 ± 10.3 mL (range, 2.8–88.0 mL). The crude prostate volume increased with age. The mean corrected prostate volume (divided by body surface area) was significantly higher in patients with atherosclerosis than in those without. However, multiple regression analysis revealed that only age influenced the corrected prostate volume. Age was the only significant influencing factor for prostate volume. We propose applying age estimation using prostate volume for forensic medicine purposes. Because prostate volume was not influenced by concomitant disease, it would be valuable to estimate the decedent’s age using the prostate volume.

## 1. Introduction

Because decedents undergoing forensic autopsies would have behaved normally prior to death, we can accurately confirm the normal prostate volume using forensic autopsy materials. Generally, in a young adult male, the prostate weight is approximately 10 to 15 g and increases to 30 g by the age of 50 years. Previous studies have concluded that prostate volume increases with age [1,2,3,4,5,6,7,8,9,10,11,12]. However, several studies had small sample sizes [1,2,4,5,12], limited age groups [4,8,9,11], or indirect measurements other than autopsies [8,9,10,11]. Therefore, the confirmation of the prostate volume of healthy persons is required. Several methods have been applied to examine prostate volume, such as digital rectal examination, computed tomography (CT) [13], magnetic resonance imaging (MRI) [13], transrectal ultrasonography [8,9,11,13,14,15,16], and autopsy [1,2,3,4,5,6,7]. Of these, the prostate volume determined using autopsy materials is considered the most accurate data because the method involves direct measurement. Several reports have described analysis of prostate volume with autopsy materials [1,2,3,4,5,6,7]. Most of these studies were based on medical autopsies in which the decedent had been previously diagnosed with other diseases, such as malignant or chronic diseases, and had been prescribed medicines. Therefore, medical intervention or the diseased state could have influenced prostate volume. In contrast, many decedents undergoing forensic autopsies have died suddenly as a result of acute disease or external causes, such as an accident, suicide, or homicide. Thus, decedents in these scenarios would have likely behaved normally prior to death and are less likely to have been affected by long-term medical interventions. Therefore, it is possible to obtain a more accurate prostate volume in these individuals using forensic autopsy materials.

Additionally, non-communicable diseases have recently become a major issue worldwide. According to the World Health Organization, non-communicable diseases kill 41 million people each year, equivalent to 71% of all deaths globally. Metabolic syndrome, including obesity, hyperlipidemia, and diabetes mellitus (DM), has become a frequently occurring disease in developed countries, including Japan. Previous studies have suggested that some of these diseases can influence the prostate volume [10,11,13,15,16,17]. However, no report has confirmed the effect of the concomitant disease on prostate volume using a forensic autopsy. Because the presence or previous history of any disease can be obtained at forensic autopsy and the subsequent histopathological examination, we can accurately evaluate the health status of a decedent and confirm whether the disease influenced their prostate volume.

The objectives of this study were to first confirm the current prostate volume by age and then determine diseases that can influence prostate volume using forensic autopsy materials.

## 2. Materials and Methods

### 2.1. Specimens

Prostate specimens were collected from forensic autopsies performed at Shiga University of Medical Science, Japan, between January 2015 and December 2019. All forensic autopsies in Shiga Prefecture were performed at this university. Of the 633 decedents autopsied over the period, 445 males were chosen. Bodies that were severely affected by trauma or decomposition that caused difficulty in removing the entire prostate were excluded. Furthermore, decedents who were under 21 years old or lacked any information were also excluded. Finally, 207 decedents were selected as the subjects. General information, including age, height, weight, body mass index, and past medical history, was obtained. The body surface area of each individual was calculated according to the height and weight with a previously reported method [18]. Each patient’s past medical history was reviewed through interviews with their family and referring physicians, if possible. No decedents in this series had undergone transurethral resection of the prostate or transurethral enucleation with bipolar energy. Because forensic autopsies routinely include pathological examinations of the entire body, prostates are usually examined both macroscopically and microscopically. Therefore, the collection, processing, and examination of the prostate were not beyond routine forensic autopsy protocol.

This study was approved by the Ethics Committee of Shiga University of Medical Science (No. 16–22).

### 2.2. Measurement of Prostate Volume

After removing the prostate, we removed its surrounding soft tissue completely. Then, the prostate was put in a water-filled beaker. The overflowed water volume was considered as the prostate volume (Archimedes’ principle).

### 2.3. Concomitant Disease

We examined the concomitant diseases in each individual. From previous histories, we determined whether the individual had suffered from hypertension, heart disease, or cerebrovascular disease. Similarly, we examined the autopsy records to determine if any of these diseases were present. Atherosclerosis was defined in this study as having a calcification or ulceration lesion in the aorta at autopsy.

### 2.4. Statistical Analysis

To compare the mean corrected prostate volume between the two groups, an F-test was first performed to examine the homogeneity of variance. An unpaired *t*-test was used for equal variance, and a Welch’s *t*-test was used for unequal variances. Multiple regression analysis using the forced input method was performed with the corrected prostate volume as the target variable and age and the presence of atherosclerosis, hypertension, heart disease, and cerebrovascular disease as descriptive variables. The statistical analyses were performed with IBM SPSS version 23 (IBM Corp., Armonk, NY, USA).

## 3. Results

### 3.1. General Characteristics

The mean age of the decedents was 6.6 ± 18.l years, ranging from 21 to 92 years. The age distribution of the decedents is shown in Figure 1. In terms of postmortem duration, 59 cases (28.5%) were within 24 h and 84 cases were between 24 and 48 h. The remaining 64 cases were above 48 h, up to one month. In terms of cause of death, 72 (35%) individuals died suddenly of disease, and 135 deaths (65%) involved external causes. Because the prostate volume was not significantly different between the deaths from disease and external causes (*p* = 0.559), the following analyses were performed for all cases.

### 3.2. Volume of the Prostate

The distribution of the crude prostate volume is shown in Figure 2. The mean crude prostate volume was 29.1 ± 10.3 mL, ranging from 2.8 mL to 88.0 mL. The mean crude prostate volume in each age group is shown in Table 1. The mean crude prostate volume according to the postmortem duration was 28.8 ± 11.1 mL in decedents with a postmortem duration of ≤48 h and 29.5 ± 8.8 mL in decedents with a postmortem duration of >48 h, and the difference was not statistically significant (*p* = 0.627).

Next, we compared the obtained crude prostate volume of each age group with the previous studies. The results of comparisons to previous results measured at autopsy [7] are shown in Figure 3A, and the results of comparisons to previous results measured via ultrasonography [8,9] are shown in Figure 3B. In both comparisons, the crude prostate volume increased with age.

### 3.3. Prostate Volume and Diseases

We then examined the effect of concomitant diseases on prostate volume. Because prostate volume was influenced by body size [11,13], the corrected prostate volume was defined as the crude prostate volume divided by the body surface area (mL/m^2^). The mean corrected prostate volume was compared between those with concomitant disease and those without (Table 2). Among the four diseases examined, the decedents suffering from atherosclerosis had a significantly higher corrected prostate volume than those without.

### 3.4. Factors Influencing the Prostate Volume

Table 3 shows the results of the multiple regression analysis for the corrected prostate volume. The results suggest that only the decedent age independently influenced the corrected prostate volume (*p* < 0.001).

## 4. Discussion

In this study, we clarified the prostate volume of healthy persons using forensic autopsy materials. Previously, the average prostate weight was presented for a 10-year interval from the 925-autopsy specimens included in five studies [7]. For the prostate growth rate, the weight gradually increased from the age of 21 to 70 years, but then more markedly increased after the age of 71. This trend was similar to the current results here. When comparing the volume in aged victims, the present results showed similar values to the previous combined data [7]. Although these previous data only included a few samples (four specimens of individuals between 71 and 80 years old and two specimens of individuals 81 years old or more [5]), our results confirmed the prostate volume of aged persons.

When comparing the present prostate size data of decedents aged 21 to 70 years with the previous combined data, our results showed slightly larger values. This is possibly because of the differences between medical autopsy and forensic autopsy specimens or the lack of direct measurement of the total volume or weight [1,2]. Disparities in prostate weight between medical autopsy and forensic autopsy specimens were reported previously. Leissner and Tisell found that decedents who had suffered from prolonged and wasting disease had lower prostate weights (as a result of reduced Leydig cell weights following protracted illness before death [19]) than those who had died suddenly or after a disease of short duration [6].

The prostate volume was also determined via ultrasonography [8,9]. When comparing the present values of decedents aged 31 to 50 years with those of normal prostates determined via ultrasonography, similar results were obtained (mean of 25.5 vs. 23.9 in those aged 31 to 40 years; 27.4 vs. 25.7 in those aged 41 to 50 years) [8,9]. However, in older decedents, our values were smaller than those derived via ultrasonography. Specifically, our mean values were 28.7 in those aged 65 to 69 years and 30.7 in those aged 70 to 74 years, while those derived via ultrasonography were 37.9 and 44.9, respectively. Although the reasons for these differences were not determined scientifically, we consider that they may have been due to the blood flow (most blood volume in the prostate is lost in autopsy materials) or technical issues of ultrasonography, such as the imaging quality or the ultrasonographers’ personal experience with prostate volume measurements. Because a larger prostate has more circulating blood, the differences between the values of a living person compared with an autopsy specimen may be more apparent in older individuals. Additionally, these differences may be because a normal prostate in a young man exhibits a triangular-like shape on transverse section, but it becomes rounded and ellipsoid in an age-dependent manner. Therefore, volume estimation via ultrasonography might be inaccurate in older people [9,20,21]. Our prostate volume data in each age group had the highest reliability, as it was measured directly using the Archimedes’ principle and using forensic autopsy specimens.

According to the present study, age was a significant positive factor for prostate volume, but the presence of atherosclerosis, hypertension, cerebrovascular disease, or heart disease was not. A meta-analysis showed that prostate volume was significantly higher in older, obese patients and patients with low serum high-density lipoprotein (HDL) cholesterol concentrations [10]. Another study that examined the impact of metabolic component and body composition indices on prostate volume in middle-aged men undergoing health check-ups suggested that age was a significant predictor, but raised blood pressure, raised fasting blood sugar, raised triglyceride, and reduced HDL levels were not [11]. The influence of age shown in our study is in accordance with these previous results. Because high blood pressure, raised fasting blood sugar, and reduced HDL are risk factors of cerebrovascular or cardiovascular disease, our results of no significant effects of these diseases are also consistent with these studies.

A retrospective analysis of patients who underwent transurethral resection of the prostate suggested that a number of cardiovascular risk factors, such as hypertension, DM, smoking, and dyslipidemia, did not influence the prostate volume [17]. However, in the Pearson correlation analysis, both systolic and diastolic blood pressure were significantly related to prostate volume. Because this study examined patients with lower urinary tract symptoms or benign prostate hypertrophy, the results are different from our findings. Hypertension can possibly cause vascular damage, leading to increased flow resistance, subsequently triggering prostatic growth [16]. No significant difference was found after comparing the prostate volume among patients with DM, peripheral arterial occlusive disease, coronary artery disease, or controls [17]. This was in accordance with our result that atherosclerosis was not a significant factor influencing prostate volume. Because the relationship between each factor and prostate volume depended on the background of the patients or victims, such as patients with benign prostate hypertrophy, an average person, or an aged person, the results may be varied to some extent.

Our data confirm that among age and the presence of atherosclerosis, hypertension, heart disease, or cerebrovascular disease, age was the only significant influencing factor for prostate volume. Therefore, we propose applying age estimation using prostate volume from a forensic medicine perspective. Because prostate volume was not influenced by some risk factors for metabolic syndrome or the presence of cardiovascular disease, hypertension, heart disease, or cerebrovascular disease, it would be valuable to estimate the decedent’s age by prostate volume in the future. Numerous medical imaging technologies are available to estimate the prostate volume, such as MRI, CT, abdominal ultrasound, and transrectal ultrasound. Automatic volume estimation procedures were recently developed from abdominal ultrasound images [22]. Such systems might help decrease interobserver differences that arise when using manual methods and reduce the time spent performing imaging and calculations. In future, if the estimated volumes obtained using these medical imaging technologies are confirmed by the values obtained at forensic autopsies, the quality of automatic volume estimation systems would be improved. Therefore, the present results might serve as the baseline data set for confirmation procedures. Artificial intelligence (AI) has recently been applied in the field of forensic medicine [23,24]. Conventional neural networks, which are a type of deep learning algorithm, are widely used in pattern recognition and imaging processing. AI has facilitated the use of X-ray images, MRI, photography, and CT scans of the head or other bones (e.g., collarbone, femur, and teeth) for age estimation [23,24]. In addition to the reported data set for age estimation, we propose the application of the prostate volume obtained via imaging techniques for the development of a conventional neural network. We believe that the present results will also contribute to the development of AI-based age estimation in future forensic medicine.

This study has several limitations. First, because the prostate specimens were collected at autopsy, cases with latent prostate carcinoma were also included. A previous study showed that the prevalence of latent prostate carcinoma in Japan increased with age, reaching 23.6% in the seventh decade of life and 33.3% in the eighth decade [25]. Therefore, the presence of latent prostate carcinoma may influence the volume of the prostate. Our objective was to confirm the prostate volume of people in general, so we used all cases for analyses. In future studies, we will divide the cases into those with prostatic diseases and those without, then compare their volumes. Second, the sample set included cases with relatively long postmortem durations. From our experience, we considered that the influence of postmortem duration on prostate volume was not significant if the body was not heavily decomposed. Furthermore, our data revealed no significant difference in the mean crude prostate volume according to the postmortem duration. However, in the future, only the samples with short postmortem durations will be used for a similar analysis to confirm the present results. Third, some information, such as previous medical history, was obtained from the families or police investigations. Therefore, some information might have been lost.

## 5. Conclusions

Our results confirmed that prostate volume was not influenced by some risk factors for metabolic syndrome or the presence of cardiovascular disease (atherosclerosis), hypertension, heart disease, or cerebrovascular disease; however, prostate volume was significantly influenced by age. Therefore, we propose the use of prostate volume as a means of age estimation in the forensic medicine context. We believe that the present results will also contribute to the development of AI-based age estimation in future forensic medicine.

## Figures and Tables

**Figure 1 healthcare-11-01486-f001:**
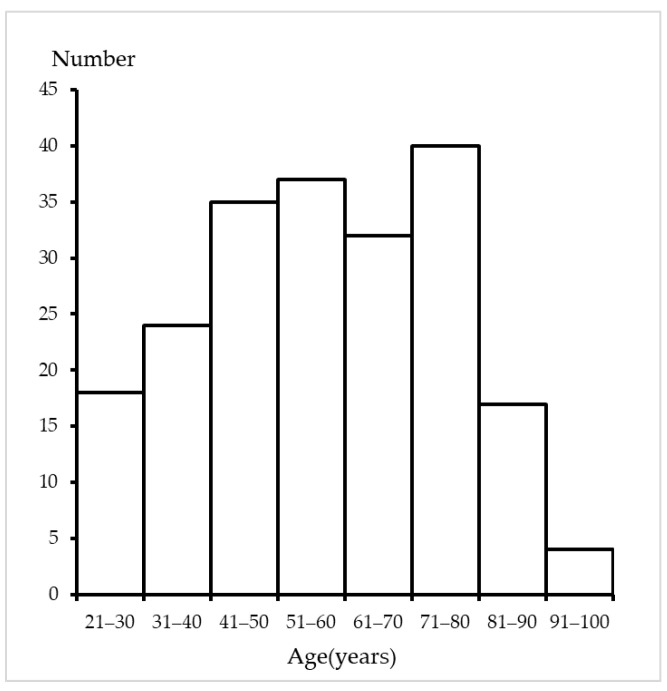
Distribution of decedent age (years).

**Figure 2 healthcare-11-01486-f002:**
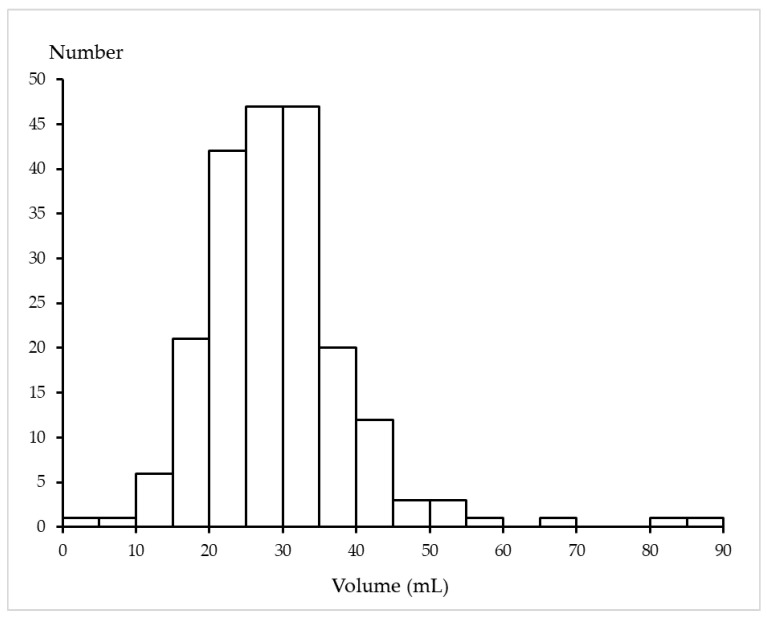
Distribution of the crude prostate volume (mL).

**Figure 3 healthcare-11-01486-f003:**
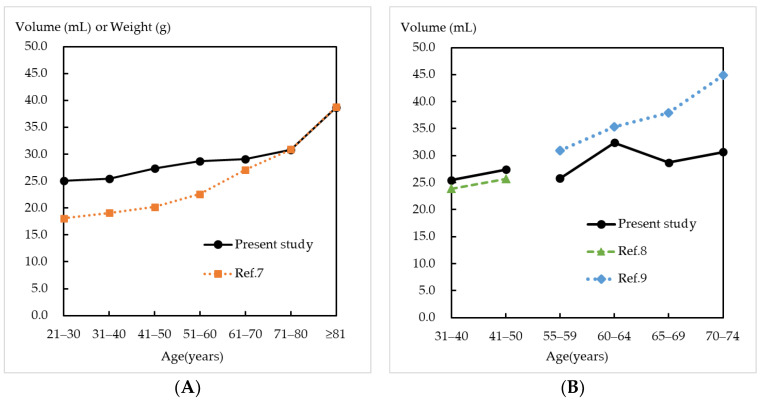
Comparisons to the previous results [7] measured at autopsy (**A**) and to the previous results [8,9] measured via ultrasonography (**B**).

**Table 1 healthcare-11-01486-t001:** Mean crude prostate volume in each age group.

Age group	21–30	31–40	41–50	51–60	61–70	71–80	≥81
Mean prostate volume (mL)	25.1	25.5	27.4	28.7	29.1	30.8	38.7
Standard deviation	7.0	6.4	8.4	6.7	7.9	8.3	21.2

**Table 2 healthcare-11-01486-t002:** The effect of concomitant diseases on prostate volume.

Previous History	+		−		*p*
	*n*	Corr. Vol. (mL/m^2^)	*n*	Corr. Vol. (mL/m^2^)	
Atherosclerosis	127	18.2 ± 7.5	80	15.1 ± 4.1	<0.001
Hypertension	45	17.9 ± 5.1	162	16.7 ± 6.9	0.211
Heart disease	92	17.8 ± 7.3	115	16.4 ± 5.9	0.145
Cerebrovascular disease	27	17.2 ± 3.9	180	17.0 ± 6.9	0.819

Corr. Vol.: corrected volume.

**Table 3 healthcare-11-01486-t003:** Results of the multiple regression analysis.

Factors	*β*	*p*
Age	0.492	<0.001
Atherosclerosis	−0.056	0.521
Hypertension	−0.033	0.638
Heart disease	−0.048	0.497
Cerebrovascular disease	−0.068	0.333

## Data Availability

The data presented in this study are available upon request from the corresponding author.

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
