# Peer review of "Affecting Factors of Prostate Volume in Forensic Autopsied Decedents"

_healthcare, 2023, doi:10.3390/healthcare11101486_

Round 1
Reviewer 1 Report
The manuscript titled "Affecting factors of prostate volume in forensic autopsied decedents" submitted to Healthcare journal (ID: healthcare-2330819) focuses on the correlation of prostate volume with age and chronic metabolic diseases, such as atherosclerosis, diabetes, etc. Data were collected from 207 autopsies performed between January 2015 and December 2019. the results excluded any correlation between the aforementioned metabolic diseases and lay the foundations for future and more accurate studies on prostate volume.
The study is well-structured, and the design seems appropriately carried out. However, a few points need clarification:
Regarding the Introduction and in particular line n. 28, I find the following sentence irrelevant to the object of the study: "Understanding the normal prostate volume is important for diagnosing benign hyperplasia and natural histories relating to the prostate", as the article does not compare healthy prostate volumes from "pathological" ones. Therefore, I would suggest removing this part, as everything appears more focused on volumetric measurement with reference to chronic diseases and age.
In line n.31, the Authors state that "Numerous studies have concluded that the prostate volume increases with age.", highlighting the lack of novelty of the manuscript, as the results can be superimposed to previously published studies.
The Materials and methods section is clearly presented, as the study was also approved by the "Ethics Committee of Shiga University of Medical Science", as stated in line 77.
From line 153 to line 162 of the Discussion begins a digression on the disparity of volumetric results obtained by ultrasound on live patients compared to the autopsy examination, due to the lack of blood circulation in the deceased subject which would lead to a variation in the value of the prostate volume, which surely needs to be kept in mind. However, this finding does not appear to be well contextualized, nor explored in relation to the results obtained.
Finally, the Conclusions of the study comment on results and considerations which, as previously mentioned, are substantially in line with the literature and other scientific articles, i.e. they do not correlate prostate volume increases with metabolic diseases such as cardiovascular diseases, identifying in age the only element of direct correlation with the increase in prostate volume.
Basically, although the article is well structured overall, I do not think the study itself brings anything innovative to the scientific literature. Therefore, I would suggest against publication, as the study needs to be implemented with further research elements.
Author Response
To Reviewer 1:
Thank you for your thoughtful and constructive feedback on our manuscript. We have revised the manuscript in accordance with your suggestions. We are grateful for the time and energy you have spent on our behalf.
The study is well-structured, and the design seems appropriately carried out. However, a few points need clarification:
Regarding the Introduction and in particular line n. 28, I find the following sentence irrelevant to the object of the study: "Understanding the normal prostate volume is important for diagnosing benign hyperplasia and natural histories relating to the prostate", as the article does not compare healthy prostate volumes from "pathological" ones. Therefore, I would suggest removing this part, as everything appears more focused on volumetric measurement with reference to chronic diseases and age.
We have accordingly removed the sentence.
In line n.31, the Authors state that "Numerous studies have concluded that the prostate volume increases with age.", highlighting the lack of novelty of the manuscript, as the results can be superimposed to previously published studies.
We have accordingly revised the above-mentioned sentence in the Introduction section as follows (Lines 32–36): “Previous studies have concluded that prostate volume increases with age [1-12]. How-ever, several studies had small sample sizes [1, 2, 4, 5, 12], limited age groups [4, 8, 9, 11], or indirect measurement other than autopsies [8-11]. Therefore, confirmation of the prostate volume of healthy persons is required.”
The Materials and methods section is clearly presented, as the study was also approved by the “Ethics Committee of Shiga University of Medical Science”, as stated in line 77.
Thank you for the positive evaluation.
From line 153 to line 162 of the Discussion begins a digression on the disparity of volumetric results obtained by ultrasound on live patients compared to the autopsy examination, due to the lack of blood circulation in the deceased subject which would lead to a variation in the value of the prostate volume, which surely needs to be kept in mind. However, this finding does not appear to be well contextualized, nor explored in relation to the results obtained.
According to your comment, we have revised the text in the Discussion section as follows (Lines 171–175): “Although the reasons for these differences were not determined scientifically, we con-sider that may be due to the blood flow (most blood volume in the prostate are lost in autopsy materials) or technical issues of ultrasonography, such as the imaging quality or the ultrasonographers’ personal experience with prostate volume measurements.”
Finally, the Conclusions of the study comment on results and considerations which, as previously mentioned, are substantially in line with the literature and other scientific articles, i.e. they do not correlate prostate volume increases with metabolic diseases such as cardiovascular diseases, identifying in age the only element of direct correlation with the increase in prostate volume.
Basically, although the article is well structured overall, I do not think the study itself brings anything innovative to the scientific literature. Therefore, I would suggest against publication, as the study needs to be implemented with further research elements.
We believe that the present results might contribute to the development of age estimation in future forensic medicine. Artificial intelligence (AI) has recently been applied in the field of forensic medicine, especially for the practice of age estimation [Healthcare 2021, 9, 1545]. AI has allowed X-ray images, magnetic resonance images, photography, and computed tomography scans of the head or other bones (e.g., collarbone, femur, teeth) to be used for age estimation. Because the qualities of prostate volume estimation by these medical imaging technologies would be improved by our results [Applied Science 2022, 12, 1390], we propose application of the prostate volume obtained by imaging techniques for AI-based age estimation. Further extensive research regarding the prostate volume and age may be published based on the present results.
In accordance with your comment, we have added the following text to the last two paragraphs of the Discussion section along with three more references (Lines 216–233): “Numerous medical imaging technologies are available to estimate the prostate volume, such as MRI, CT, abdominal ultrasound, and transrectal ultrasound. Automatic volume estimation procedures were recently developed from abdominal ultrasound images [22]. Such systems might help decrease interobserver differences that arise by manual methods and reduce the time spent performing imaging and calculations. In future, if the estimated volumes obtained by these medical imaging technologies are confirmed by the values obtained at forensic autopsies, the quality of automatic volume estimation systems would be improved. Therefore, the present results might serve as the baseline data set for confirmation procedures. Artificial intelligence (AI) has recently been ap-plied in the field of forensic medicine [23, 24]. Conventional neural networks, which are a type of deep learning algorithm, are widely used in pattern recognition and imaging processing. AI has facilitated the use of X-ray images, MRI, photography, and CT scans of the head or other bones (e.g., collarbone, femur, and teeth) for age estimation [23, 24]. In addition to the reported data set for age estimation, we propose application of the prostate volume obtained by imaging techniques for development of a conventional neural network. We believe that the present results will also contribute to the development of AI-based age estimation in future forensic medicine.”
Additionally, the following sentence has been added to the end of the Conclusion section (Lines 255–256): “We believe that the present results will also contribute to the development of AI-based age estimation in future forensic medicine.”

Reviewer 2 Report
Dear Authors,
I have read with interest the manuscript 'Affecting factors of prostate volume in forensic autopsied decedents'. Thank you for the opportunity to review your manuscript!
The premise is excellent but there are some issues that must be clarified.
Point 1: Please split the abstract according the Instructions for authors of Healthcare:
The abstract should be a single paragraph and should follow the style of structured abstracts, but without headings: 1) Background: Place the question addressed in a broad context and highlight the purpose of the study; 2) Methods: Describe briefly the main methods or treatments applied. Include any relevant preregistration numbers, and species and strains of any animals used. 3) Results: Summarize the article's main findings; and 4) Conclusion: Indicate the main conclusions or interpretations. The abstract should be an objective representation of the article: it must not contain results which are not presented and substantiated in the main text and should not exaggerate the main conclusions.
Point 2: Materials and Methods, Paragragh 2.1. In line 65 you describe as exclusion criteria the decomposition. When you describe the range of post-mortem interval it is clear that in about half of your sample death occurred from more than 48 hours up to one month. You report this between the limits of the study, but you should in any case perform an analysis in order to infer that "the influence of postmortem duration on prostate volume was not significant if the body was not heavily decomposed". Please show it in the results.
Point 3: Statistical analysis.
Please provide information about the Software you used for the analysis.
Point 4: Have you excluded specimen coming from people with previous TURP? In case, underline it in matherials and methods.
Point 5: Results: may you provide a table with the mean prostrate volumes stratified per age groups?
Point 6: in the conclusions you set as a limit of the study the presence of prostate carcinoma. This may be a limit but you should also consider that in elderly and particularly in over 70 y old people the incidence of prostate cancer is near 90% of the population. This reflects obviously that the volume is related with the age. Please explain it in the discussion.
Author Response
To Reviewer 2:
Thank you for your thoughtful and constructive feedback on our manuscript. We have revised the manuscript in accordance with your suggestions. We are grateful for the time and energy you have spent on our behalf.
The premise is excellent but there are some issues that must be clarified.
Point 1: Please split the abstract according the Instructions for authors of Healthcare:
The abstract should be a single paragraph and should follow the style of structured abstracts, but without headings: 1) Background: Place the question addressed in a broad context and highlight the purpose of the study; 2) Methods: Describe briefly the main methods or treatments applied. Include any relevant preregistration numbers, and species and strains of any animals used. 3) Results: Summarize the article's main findings; and 4) Conclusion: Indicate the main conclusions or interpretations. The abstract should be an objective representation of the article: it must not contain results which are not presented and substantiated in the main text and should not exaggerate the main conclusions.
In accordance with your comment, we have revised the first sentence of the abstract to give a background of the study as follows (Lines 10–11): “Because decedents undergoing forensic autopsies would have behaved normally before death, prostate volume according to age group can be confirmed with forensic autopsy materials.” We have not added headings to the abstract; it has been formatted as a single paragraph without headings in accordance with the instructions to authors and your comment above.
Point 2: Materials and Methods, Paragragh 2.1. In line 65 you describe as exclusion criteria the decomposition. When you describe the range of post-mortem interval it is clear that in about half of your sample death occurred from more than 48 hours up to one month. You report this between the limits of the study, but you should in any case perform an analysis in order to infer that "the influence of postmortem duration on prostate volume was not significant if the body was not heavily decomposed". Please show it in the results.
To examine the influence of the postmortem duration on prostate volume, we compared the prostate volume between the decedents with a postmortem duration of ≤48 hours and those with a postmortem duration of >48 hours. However, the values were similar (28.8 ± 11.1 vs. 29.5 ± 8.8 mL) and showed no significant difference. Therefore, in accordance with your suggestion, we have added the following text to the Results section (Lines 115–119): “The mean crude prostate volume according to the postmortem duration was 28.8 ± 11.1 mL in decedents with a postmortem duration of ≤48 hours and 29.5 ± 8.8 mL in decedents with a post-mortem duration of >48 hours, and the difference was not statistically significant (P = 0.627).”
We have also added the following sentence to the limitations paragraph in the Discussion section (Lines 244–245): “Furthermore, our data revealed no significant difference in the mean crude prostate volume according to the postmortem duration.”
Point 3: Statistical analysis.
Please provide information about the Software you used for the analysis.
We have accordingly added the following sentence to the Statistical analysis section (Lines 100–101): “The statistical analyses were performed with IBM SPSS version 23 (IBM Corp., Armonk, NY, USA).”
Point 4: Have you excluded specimen coming from people with previous TURP? In case, underline it in matherials and methods.
No decedents underwent transurethral resection of the prostate or transurethral enucleation with bipolar energy. In accordance with your suggestion, we have added the following sentence to the Material and Methods section (Lines 77–78): “No decedents in this series had undergone transurethral resection of the prostate or transurethral enucleation with bipolar energy.”
Point 5: Results: may you provide a table with the mean prostrate volumes stratified per age groups?
We have accordingly added a table (new Table 1) to the revised manuscript and the following sentence to the Results section (Lines 115–116): “The mean crude prostate volume in each age group is shown in Table 1.” As a result, the table numbers have changed.
Point 6: in the conclusions you set as a limit of the study the presence of prostate carcinoma. This may be a limit but you should also consider that in elderly and particularly in over 70 y old people the incidence of prostate cancer is near 90% of the population. This reflects obviously that the volume is related with the age. Please explain it in the discussion.
We have accordingly added the following text with an additional reference to the limitations paragraph (Lines 235–239): “A previous study showed that the prevalence of latent prostate carcinoma in Japan in-creased with age, reaching 23.6% in the seventh decade of life and 33.3% in the eighth decade [25]. Therefore, the presence of latent prostate carcinoma may influence the volume of the prostate.”

Round 2
Reviewer 1 Report
The manuscript has been improved, and the suggested modifications have been applied. The novelty of the study is quite average; however, it is surely fit for publication in the present form.
Reviewer 2 Report
Dear Authors,
I have appreciated the new form of the manuscript, and that you have taken into account the suggestions.
It can be a good piece of scholarship!